# Comparative efficacy and acceptability of non-pharmacological interventions for depression among people living with HIV: A protocol for a systematic review and network meta-analysis

**Ting Zhao**[1,2]**, Chulei Tang**[3]**, Huang Yan**[1,2,4]**, Honghong Wang**[1,2]*****, Meiying Guo**[4]

**1** Xiangya School of Nursing, Central South University, Changsha, Hunan, China, **2** Xiangya Center for Evidence-Based Nursing Practice & Healthcare Innovation: A Joanna Briggs Institute Affiliated Group, Changsha, Hunan, China, **3** School of Nursing, Nanjing Medical University, Nanjing, Jiangsu, China, **4** Nursing Department, The Third Xiangya Hospital, Central South University, Changsha, Hunan, China

* honghong_wang@hotmail.com

## Abstract

### Background

Improving depression is critical to the success of HIV treatment. Concerns about the adverse effects of pharmacotherapy have led to non-pharmacological treatments for depression in people living with HIV (PLWH) becoming increasingly popular. However, the most effective and acceptable non-pharmacological treatments for depression in PLWH have not yet been determined. This protocol for a systematic review and network meta-analysis aims to compare and rank all available non-pharmacological treatments for depression in PLWH in the global network of countries as well as in the network of low-income and middle-income countries (LMICs) only.

### Methods

We will include all randomized controlled trials of any non-pharmacological treatments for depression in PLWH. The primary outcomes will consider efficacy (the overall mean change scores in depression) and acceptability (all-cause discontinuation). Published and unpublished studies will be systematically searched through the relevant databases (PubMed, EMBASE, Cochrane Central Register of Controlled Trials (CENTRAL), PsycINFO, CINAHL, ProQuest, and OpenGrey), international trial registers, and websites. There is no restriction by language and publication year. All study selection, quality evaluation, and data extraction will be independently conducted by at least two investigators. We will perform a random-effects network meta-analysis to synthesize all available evidence for each outcome and obtain a comprehensive ranking of all treatments for the global network of countries as well as for the network of LMICs only. We will employ validated global and local approaches to evaluate inconsistency. We will use OpenBUGS (version 3.2.3) software to fit our model within a Bayesian framework. We will evaluate the strength of evidence using the

**Data Availability Statement:** No datasets were generated or analysed during the current study. All

relevant data from this study will be made available upon study completion.

**Funding:** This work was supported by the Provincial Natural Science Foundation of Hunan Grant, China (2022JJ30769). The funders did not and will not have a role in study design, data collection and analysis, decision to publish, or preparation of the manuscript.

**Competing interests:** The authors have declared that no competing interests exist.

Confidence in Network Meta-Analysis (CINeMA) tool, a web application based on the Grading of Recommendations, Assessment, Development and Evaluation (GRADE) system.

## Ethics and dissemination

This study will use secondary data and therefore does not require ethical approval. The results of this study will be disseminated through peer-reviewed publication.

## Trial registration

PROSPERO registration number: CRD42021244230.

## Introduction

According to the World Health Organization, 38.4 million people are currently living with HIV at the end of 2021, and 650,000 people died from HIV-related causes in 2021 [1]. As a life-altering disease associated with social stigma, comorbidities, adverse effects related to anti-retroviral therapy, and uncertainty of prognosis, people living with HIV (PLWH) commonly experience psychological disorders [2–5]. The latest statistical analyses indicate that approximately 22%–44% of PLWH worldwide suffer from depression [6], which significantly contributes to poor HIV treatment outcomes, such as worse adherence to treatment, poor viral suppression, accelerated disease progression, suicidal ideation, and so on [7–11].

Numerous pharmacological and non-pharmacological treatments have been proposed for depression in PLWH. Although pharmacotherapy is a cornerstone of depression management in PLWH, there are concerns about potential adverse effects (e.g., somatic distress, sexual dysfunction, drug abuse, and lethality in overdose), drug interactions with HIV medications, and withdrawal and rebound phenomena [12–17]. This has caused non-pharmacological treatments for depression in PLWH to become increasingly popular.

There is a wide choice of non-pharmacological treatments, including psychotherapy (e.g., cognitive behavioral therapy) [18, 19], psychosocial treatments (e.g., meditation and yoga) [20–22], physical therapies (e.g., exercise) [23–26], music therapy [27], and so on. Among them, clinical practice guidelines recommend psychotherapy as the first-line treatment for depression [28, 29]. However, such treatments rely on mental health professionals for delivery. In most countries, particularly low-income and middle-income countries (LMICs), the limited availability and accessibility of psychotherapy resources restrict the application of mental health services [30, 31]. Evidence suggests that the rate of available mental health therapists per 100,000 people in LMICs is about 0.5% of that in high-income countries [31–33], and in some cases, only one or two psychiatrists are available for the entire country [32]. Thus, some treatments that can complement or alternate with psychotherapy remain indispensable. However, although many non-pharmacological resources have been shown to effectively alleviate depression in PLWH, these non-pharmacological treatments are rarely systematically compared. Traditional pairwise meta-analyses could investigate only the relative effect of a single direct comparison (usually a treatment compared to the usual care or waiting lists), rather than performing multiple comparisons simultaneously, and this is further limited by the lack of availability of a sufficient number of head-to-head comparisons between different treatments [34]. The most effective and acceptable non-pharmacological treatment for depression among PLWH in different resource contexts is yet to be identified by further research.

Network meta-analysis can be used to synthesize data across multiple comparisons and improve the precision of effect estimates by combining all available direct and indirect evidence simultaneously [35–37]. In addition, the relative effect estimates can be used to rank treatments and identify the optimal treatment for depression in PLWH [38]. All these measures contribute to the process of synthesizing the evidence [39]. Therefore, we aim to conduct a systematic review and network meta-analysis to compare and rank the relative efficacy and acceptability of all available non-pharmacological treatments for depression in PLWH in the global network of countries as well as in the network of LMICs only.

## Materials and methods

This systematic review protocol follows the Preferred Reporting Items for Systematic Review and Meta-analysis Protocols (PRISMA-P) 2015 statement. The study has been registered with PROSPERO (CRD42021244230).

### Eligibility criteria

**Types of studies.**   Any randomized controlled trials (RCTs) of non-pharmacological interventions for depression in PLWH will be included, including cross-over trials and cluster randomized trials. We will exclude quasi-randomized trials.

**Types of participants.**   We will include RCTs in which sampled adults living with HIV have met standardized diagnostic criteria for at least mild depression or have been identified as suffering from at least mild depression at baseline data based on any validated rating scale for depression. To reduce clinical heterogeneity, we will exclude RCTs involving 20% or more participants with bipolar depression, treatment-resistant depression, psychotic depression, or peripartum depression, but not studies involving participants with other comorbid psychiatric disorders (e.g., anxiety disorder). We will also exclude studies in which participants have serious concomitant medical illnesses other than HIV

**Types of interventions.**   We will only take into account any non-pharmacological interventions recommended by current guidelines. To reduce inconsistency among trials, we will exclude studies that combined non-pharmacological interventions with specific drugs and studies using specific drugs as the comparator. We will consider different intensity levels or subtypes within the same type of non-pharmacological intervention as the same node in the network analysis in the first instance or as different nodes if adequate data are available. All the eligible interventions will constitute a network of interventions. In principle, all eligible participants are equally likely to be randomly assigned to any interventions in the network.

**Types of outcome measures.**   *Primary outcomes.*

1. Efficacy (continuous variable)–measured by the overall mean change scores (from baseline to post-intervention) on the standardized and validated depressive symptom scales, such as the Hamilton Depression Rating Scale (HAMD), [40] the Beck Depression Inventory I or II (BDI) [41, 42], the Center for Epidemiologic Studies Depression Scale (CES-D) [43], and so on. We will include studies with depression as a primary or secondary outcome, but we will exclude scores that combined depression and other symptoms. If the researchers have used more than one depression scale simultaneously to measure depression scores in a study, we will apply a pre-defined hierarchy to extract the most appropriate data. This hierarchy will be based on the evidence from systematic reviews (Table 1) [44–48]. For scales not included in this table, we will follow the following rules: (1) prefer clinician-reported scales rather than self-reports; (2) prioritize scales specifically targeted to measure depression symptoms over scales with a broader scope; (3) prefer the most commonly reported scale across studies [49].

2. Acceptability–all-cause discontinuation (dichotomous variable), defined as the proportion of participants who have discontinued treatment for any reason by the post-intervention time point [50, 51].

## Literature search

With the assistance of a senior librarian, a systematic search strategy has been developed. We will preliminarily search the Cochrane Database of Systematic Reviews (CDSR), PROSPERO, and the JBI database for similar ongoing and published systematic reviews/protocols to avoid research waste. Next, we will search through the databases of PubMed, EMBASE, the Cochrane Central Register of Controlled Trials (CENTRAL), PsycINFO, and CINAHL for relevant original articles. The search terms and search algorithm are available in the S1 Appendix. Additionally, we will conduct searches in ClinicalTrials.gov, the International Clinical Trials Registry Platform (ICTRP) of the World Health Organization, the EU Clinical Trials Register, ProQuest, and OpenGrey for published, unpublished, and ongoing studies. We will also search for key conference proceedings in the field conducted by the American Psychiatric Association, the International College of Neuropsychology, the International AIDS Society, and the Conference on Retroviruses and Opportunistic Infections. Additional studies will be identified by hand-checking the bibliographies of the included studies and relevant review papers. There is no restriction by language or publication year. We will conduct a further search before the final analysis.

## Study selection

All retrieved records will be imported into the Endnote X9 reference management tool for data management. After removing duplicate records, two investigators (TZ and HY) will independently screen all titles and abstracts identified in the searches. Then the same two investigators will separately access the full texts of the remaining articles for eligibility. When a study has been published in duplicate, we will include the most informative and complete one. If any disagreement occurs between the investigators, the third reviewer will provide arbitration. We will use the PRISMA 2020 flow diagram to outline the study selection process and the reasons for excluding full-text articles [52].

## Data extraction

Two investigators (TZ and HY) will independently extract the data from the included studies using a pre-prepared structured data extraction sheet in Microsoft Excel 2020. The extracted data will include the following information: (1) study characteristics, including first author name, publication year, publication journal, publication type, number of arms, total sample

**Table 1. Hierarchy of depression symptom severity measurement scales.**

| Hierarchy | Depression symptom severity measurement scales | Abbreviation |
|---|---|---|
| 1 | The Patient Health Questionnaire-9 | PHQ-9 |
| 2 | The Hamilton Depression Rating Scale | HAM-D |
| 3 | The Montgomery–Asberg Depression Rating Scale | MADRS |
| 4 | The Beck Depression Inventory | BDI |
| 5 | The Hospital Anxiety and Depression Scale | HADS |
| 6 | The Center for Epidemiologic Studies of Depression Scale | CES-D |

size, the sample size for each group, etc.; (2) participants' characteristics at baseline, including mean age, sex, country, years living with HIV, baseline $CD_4$ T cell counts, the severity of depression, the measurement tool for depression, etc.; (3) characteristics of interventions and comparator, including the type of treatment, treatment frequency, the length and number of sessions, treatment duration, etc.; and (4) outcome measures, including all-cause discontinuation, and mean change in depression scores with corresponding standard deviations (SDs).

All outcomes will be extracted at the post-intervention time-point. The results for intention-to-treat analyses will be extracted preferentially. For cross-over studies, we will extract only data from the first period because of carryover effects. For cluster randomized trials, we will extract data that account for the clustering in the results. When SDs are not reported, we will first use the standard errors (SEs), t-statistics, p values, and so on to calculate the missing SDs. We will also contact the authors of the included study up to three times via email to obtain the relevant missing data and seek clarifications, which will be regarded as unobtainable if no clarification is provided within six weeks. The two extractors will resolve any disagreement in data extraction by consulting a third reviewer.

## Risk of bias assessment

Two investigators (TZ and HY) will independently assess the risk of bias in the included studies using the Cochrane Collaboration Risk of Bias Tool [38], which includes seven domains: (1) sequence generation; (2) allocation concealment; (3) blinding of participants; (4) blinding of personnel; (5) incomplete outcome data; (6) selective reporting; and (7) other bias. For each domain, we will grade methodological quality as 'low risk,' 'unclear risk,' or 'high risk' according to the Cochrane Handbook version 6.1.0 (Collaboration, 2020) [38]. Any disparities between the two investigators will be settled by consulting the third reviewer. The authors of the studies will be contacted for further information, if necessary. We will determine an overall risk of bias for each eligible study. Studies will be classified as 'low overall risk' if none of the domains is rated as 'high risk,' or three or fewer are rated as 'unclear risk;' moderate if one is rated as 'high risk' or none is rated as 'high risk' but four or more are rated as 'unclear risk;' and otherwise as 'high overall risk' [53]. We will provide a summary table of the risk of bias for each eligible study.

## Data synthesis

**Characteristics of included studies and information flow in the network.**   We will first generate the descriptive statistics and study population characteristics across all included studies. Network diagrams will be conducted using the 'mvmeta' packages in Stata (version 15.1) software to present the available evidence and describe the structure of the network. The size of the nodes in the network diagram will represent the corresponding sample size of each non-pharmacological intervention. The thickness of the lines will reflect the number of studies directly compared. To understand the most influential comparisons in the network and the way in which direct and indirect evidence affects the final summary data, the contribution matrix will be used to describe the percentage contribution of each direct comparison to the entire body of evidence [54].

**Standard pairwise meta-analyses and network meta-analyses.**   All data will be double-entered into the database to ensure accuracy. We will initially perform a series of conventional pairwise meta-analyses to compare all available direct evidence. We will synthesize data to obtain the summary standardized mean differences (SMDs) for continuous outcomes and the summary odds ratios (ORs) for dichotomous outcomes, both with corresponding 95% confidence intervals (95% CI). Heterogeneity will be evaluated using Cochran's Q test and $I^2$.

$P<0.10$ means the heterogeneity is statistically significant. $I^2$ reflects the level of heterogeneity. If $I^2 >50\%$, which indicates substantial heterogeneity, we will consider subgroup analysis to explore any identified heterogeneity. All pairwise meta-analyses will be conducted with Stata (version 15.1) software.

When standard pairwise meta-analysis is completed, we will then employ the network meta-analysis to synthesize all available direct and indirect evidence for each outcome. We will use arm-based data and do our analyses within a Bayesian framework using the Markov chain Monte Carlo (MCMC) methods [34]. We will use the normal likelihood for continuous outcomes and the binomial likelihood for dichotomous outcomes. For the correlations derived from multi-arm studies, we will use multivariate distributions to illustrate. We will use the random-effects model in the Bayesian analyses. We will calculate the posterior mean of residual deviance to access model fit. The closer the posterior mean of residual deviance is to the number of data points, the better the model fit [55].

We will fit our models using OpenBUGS (version 3.2.3) software. Uninformative prior distributions will be used for all model parameters. We will set the OpenBUGS program to run three Markov chains simultaneously with different arbitrarily chosen initial values. Each chain will have 100,000 simulation iterations, the first 50,000 of which will be discarded as burn-in. We will ensure the convergence of models through the visual inspection of three chains and the Brooks–Gelman–Rubin diagnostic method. A league table will be used to present the relative effect and the 95% credible intervals (CrI) of each intervention. We will also show the ranking probability distribution of each intervention using the surface under the cumulative ranking curve (SUCRA) [56]. A higher SUCRA score will indicate better efficacy or acceptability. We will assume a common heterogeneity variance parameter ($\tau$) for each network. We will compare the posterior distribution of the estimated heterogeneity with its empirical distributions to account for the magnitude of the estimated heterogeneity [57]. Specifically, we will use the first and third quantiles of their predictive distributions to characterize the amount of heterogeneity as low, moderate, or high [58]. All analyses of the primary outcomes will be duplicated within a Frequentist framework using the 'netmeta' package in the R project (version 4.1.2).

## Assessment of transitivity

Transitivity is a fundamental assumption for the validity of indirect comparisons in network meta-analysis [38]. Before running the network meta-analysis, we will evaluate the transitivity assumption by comparing the distribution of clinical and methodological variables that can act as potential effect modifiers (covariates associated with intervention effects) across the different comparisons. The following variables will be considered as the potential effect modifiers: mean baseline age, sex ratio, years living with HIV, baseline $CD_4$ T cell counts, depressive severity at baseline, treatment duration, and study quality.

## Assessment of inconsistency

The assumption of consistency assumes that different sources of evidence (direct and indirect evidence) agree with each other [38]. However, this assumption may be violated in the entire network or in some parts of the network. Therefore, both global and local methods will be used to evaluate inconsistency. Specifically, we will use the design-by-treatment test [59] to assess inconsistency in the entire network. Side-splitting analysis [60] and the loop-specific method [36] will be employed to evaluate local inconsistency between the direct and indirect estimates for each comparison and within each closed loop of evidence in the network meta-analysis, respectively. If there is no significant inconsistency, we will use the consistency

model; otherwise, an inconsistency model will be used [60]. All analyses for inconsistency will be conducted using the 'network' and 'mvmeta' packages in Stata (version 15.1) software.

## Subgroup analyses and sensitivity analyses

If we find significant heterogeneity and/or inconsistency, we will conduct subgroup analyses or network meta-regression analyses to explore the possible sources using the following characteristics: (1) sample size; (2) the severity of depressive symptoms at baseline; (3) the treatment duration; (4) the risk of bias. We assume consistent relative treatment effects estimated at the covariate value 0 or at the mean covariate value and consistent regression coefficients for the treatment by covariate interaction [61]. We hypothesize that:

- Increasing sample size reduces the effect size between intervention and comparator.

- Increasing baseline depression severity increases the effect size between intervention and comparator.

- Increasing treatment duration increases the effect size between intervention and comparator.

- Increasing risk of bias increases the effect size between intervention and comparator.

Furthermore, we will conduct sensitivity analyses for our primary outcomes by excluding studies judged to be at 'high risk of bias', studies with sample size≤20, local inconsistency spots, and unpublished data to test the robustness of the results and determine whether one or several studies dominate the estimation of the summary effect size of the interventions.

## Publication bias

When ≥ 10 studies are available, a comparison-adjusted funnel plot will be used to assess the presence of small-study effects and publication bias in the network. Furthermore, Egger's test will be also used to examine the significant publication bias in the network. All analyses for publication bias will be conducted using the 'netmeta' package in the R project (version 4.1.2).

## Strength of evidence

We will determine the overall strength of evidence with the Grading of Recommendations, Assessment, Development and Evaluation (GRADE) framework for network meta-analysis [62]. The approach will start by assuming that the evidence is of high quality and then rate down the evidence based on five criteria of evidence (study limitations, imprecision, heterogeneity or inconsistency, indirectness, and publication bias). Overall, the strength of evidence could be rated on four scales: high, moderate, low, and very low. We will use the Confidence in Network Meta-Analysis (CINeMA) tool (a web application based on the GRADE framework) during the evaluation [62, 63].

## Supporting information

**S1 Appendix. Search string.**
(DOCX)

**S1 Checklist. PRISMA-P checklist.**
(DOC)

## Acknowledgments

The authors thank all the reviewers for their assistance and support.

## Author Contributions

**Conceptualization:** Chulei Tang, Honghong Wang, Meiying Guo.

**Methodology:** Ting Zhao, Chulei Tang, Huang Yan.

**Project administration:** Ting Zhao.

**Supervision:** Honghong Wang.

**Writing – original draft:** Ting Zhao.

**Writing – review & editing:** Chulei Tang, Huang Yan, Honghong Wang, Meiying Guo.

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
