## [Decision Letter · Decision Letter 0]

24 Nov 2022

PONE-D-22-17925

Comparative  efficacy and acceptability of non-pharmacological interventions for depression among people living with HIV: A protocol for a systematic review and network meta-analysis

PLOS ONE

Dear Dr. Wang,

Thank you for submitting your manuscript to PLOS ONE. After careful consideration, we feel that it has merit but does not fully meet PLOS ONE’s publication criteria as it currently stands. Therefore, we invite you to submit a revised version of the manuscript that addresses the points raised during the review process.

The reviewers suggested to revise the paper to improve the manuscript. Please pay attention to the methodological aspects. Also, please ensure that the English was edited by professional native speakers. I recommend the authors to attach the certificate of English correction. 

We look forward to receiving your revised manuscript.

Kind regards,

Kyoung-Sae Na, M.D., Ph.D.

Academic Editor

PLOS ONE

https://journals.plos.org/plosone/s/fileid=ba62/PLOSOne_formatting_sample_title_authors_affiliations.pdf.

Reviewers' comments:

Reviewer's Responses to Questions

**Comments to the Author**

1. Does the manuscript provide a valid rationale for the proposed study, with clearly identified and justified research questions?

Reviewer #1: Yes

Reviewer #2: Yes

2. Is the protocol technically sound and planned in a manner that will lead to a meaningful outcome and allow testing the stated hypotheses?

Reviewer #1: Partly

Reviewer #2: Yes

3. Is the methodology feasible and described in sufficient detail to allow the work to be replicable?

Reviewer #1: No

Reviewer #2: Yes

4. Have the authors described where all data underlying the findings will be made available when the study is complete?

Reviewer #1: No

Reviewer #2: Yes

5. Is the manuscript presented in an intelligible fashion and written in standard English?

Reviewer #1: No

Reviewer #2: No

6. Review Comments to the Author

You may also provide optional suggestions and comments to authors that they might find helpful in planning their study.

Reviewer #1: ## General comments

Thank you for the opportunity to contribute to your work through review of this protocol paper. The planned study is a systematic review, with network meta-analysis, of the effects of non-pharmacological interventions for depression, in people living with HIV. This is not my clinical area, yet the rationale for the study appears well articulated and substantive. I am pleased to read a protocol for NMA that considers the transitivity evaluation, and coherence assessment, alongside robust systematic review methods. Too few NMA heed the cardinal importance of transitivity to valid inference. Overall, I was satisfied with the stated methods for this study. I have specific questions that are intended to strengthen your work. Although I am sympathetic that you are likely writing in your non-native language, I have noted several major grammatical errors. PLoS do not copyedit so we should address these during review.

## Specific comments

*page 3, line 57*

You have defined the abbreviation for ART, yet not HAART. Can HAART be replaced with ART?

*page 4, lines 104 to 106*

Grammar. I suggest rephrasing to "This systematic review protocol follows the Preferred Reporting Items for Systematic Review and Meta-analysis Protocols (PRISMA-P) 2015 statement. The study has been registered with PROSPERO (CRD42021244230)."

*page 4, line 114*

Whilst often reasonable to exclude unpublished data, there is evidence in other fields that these data influence effect size. Is there any evidence in this field?

*page 4, line 118*

It is not clear if the stated 'identification of mild depression based on standardized diagnostic criteria' is an inclusion criterion for the trials you will sample, or is a criterion you will apply to trials (based on their baseline charac.) yourself? I suspect the former, in which case it would read more clearly to say "We will include RCTs that sampled adults (age xxx) living with HIV and met standard diagnostic criteria for (at least) mild depression"

*page 5, line 121*

I interpret your meaning by the statement 'For the clinical difference...' as indicating that you are excluding conditions that are clinically different to isolated depression. That being so, the wording is not clear and not ideal grammatically. Consider removing the 'clinical difference' statements in this paragraph. I think it then reads well for clarity and grammar.

*page 5, lines 122-124*

This is not clear. Consider separating these sentences.

*page 5, lines 133 to 134*

Firstly, please change 'category' to 'intervention' as this is clearer. Second, please justify excluding these trials? Are you not interested in the effect of dose? Either way, my experience is that often trials with 3 or more groups will test two doses of the same intervention against another intervention. You should not exclude these, because regardless of whether you combine the two doses into the same intervention node or not, the trial still contributes at least one comparison to the network.

*page 6, line 174*

Please expand these initials to include the first letter of all the names of each author?

*page 7, line 195 to 196*

Grammar. Consider rephrasing to "When SDs are not reported...., and so on to calculate the SD.'

*page 7, line 198*

How many times?

*page 7, line 201*

This is not ideal. I recommend at least including the study (if all other criteria are satisfied) and placing those lacking data in an Appendix. This helps gauge data availability biases in the field.

*page 7, line 203 to 204*

It is not clear what you will do here. How numerous are cluster trials in this field? It is common practice to exclude them from NMA.

*page 7, line 208*

'assessed' is past tense - implying you have already done this. As this is a protocol, I suggest rephrasing to "Two investigators will independently assess methodological...."

*page 8, line 234*

The SMD has substantial limitations in the NMA context. There is the acknowledged problem of interpretability - what does proportion of unit deviation mean clinically? - that is a problem in pairwise meta-analysis as well. Specific to NMA, the validity of a SMD rests solely on the assumption of a common heterogeneity variance being clearly upheld. Have you thought about these issues?

It is a good idea to use mean differences on a common scale, to which effects on different scales can be transformed. Do the clinimetric features of the outcome scales in this field permit this?

*page 8, lines 237 to 239*

A few clarifications. Heterogeneity is correctly estimated by estimating tau^2 (aka the heterogeneity variance aka the variance of the random effects distribution). Cochran's Q is not an estimate, rather it is a chi^2 test of difference between study observations and the estimated mean of the random effects distribution (the pooled effect point estimate). I^2 is not a statistic, rather it is a proportional measure of variability. A larger I^2 indicates that a larger relative proportion of the observed heterogeneity is due to true variability between studies (true heterogeneity) rather than sampling error. See Borenstein et al. *Res Synth Meth*. DOI: 10.1002/jrsm.1230

Accordingly, I recommend rephrasing these lines and replacing 'estimated' with 'evaluated', as this term encompasses estimation, testing and measures.

*page 8, line 240*

Please add '...subgroup analysis to explore any identified heterogeneity' for clarity?

*page 9, line 248, 266 to 269*

Netmeta implements a quasi-frequentist approach, not Bayesian. There are several R packages for Bayesian NMA. Please clarify which packages you will use? As well, SUCRA scores are not calculable with netmeta - rather P-scores, which are somewhat analogous. See Rucker & Schwarzer 2015 *BMC MRM*. DOI: 10.1186/s12874-015-0060-8

*page 9, line 248 to 251*

What priors will you use for the analysis of efficacy?

Is 'no prior' different to 'uninformative'?

*page 9, line 255 to 261*

This is not good practice. You should make this decision *a priori* based on clinical judgement.

An important consideration that is not mentioned is whether you will fit a coherence or incoherence model?

*page 9, line 262*

It's convention to describe this as the 'common heterogeneity variance parameter'.

*page 10, line 283*

The methods available for coherence assessment will depend on the package.

*page 10, line 296 to 297*

These models need further description. Why will you only consider sample size? Will you fit NMReg models to explore identified heterogeneity/incoherence, or regardless? What are your assumptions about the regression coefficients? See Donegan et al. *Res Synth Meth*. DOI: 10.1002/jrsm.1327

*page 10, line 311*

Tense. 'Use' is present/future tense.

Reviewer #2: Overall this is an impactful review. However, there are many grammatical errors that need to be edited.

Abstract

“We will employ a network meta-analysis to synthesize all available evidence for each outcome and obtain a comprehensive ranking of all interventions for the global network and for low-income and middle-income countries network only.” I’m not sure what network is being referenced. Also, should it be countries’ network?

Introduction

Paragraph 2, sentence 3 “Furthermore, it may cause potential somatic distress,

drug interactions with HIV medications, drug abuse, and lethality in overdose[13].”

The support is poor as it is from the introduction to a 2009 paper and not primary literature. The above sentence in the paper needs better citation given pharmacotherapy is standard of care in the general population in high income countries and generally safe. I’m not tracking with how antidepressants are causing drug abuse. Also, most SSRIs are hard to overdose on.

Paragraph 3- low cost- varies widely. Depends on who is giving it. You go on to explain why it is not low cost- requirement of trained staff and the limited access to such staff.

Paragraph 4- You are implying these non-pharmacologic interventions bring benefits. I think you are actually weighing if they bring benefit and how much

Grammatical errors- ex. Like 87-88 should be “have been provided.” Not “has been provided”.

Network meta-analysis- should have a citation.

Types of Participants.

“For the clinical difference, we will exclude RCTs in which 20% or more of the participants are suffering from bipolar or psychotic depression, but not involving patients with other comorbid psychiatric disorders (eg. anxiety disorder).” I am not sure what this is saying. Please re-word the sentence.

“We will also exclude studies in which participants with a serious concomitant medical illness. “ HIV is a serious medical illness. I think you mean other than HIV but the sentence should be reworded.

Types of Intervention

Synthetic comparison set- needs a definition

Table 1 has inconsistent capitalization.

7. PLOS authors have the option to publish the peer review history of their article (what does this mean?). If published, this will include your full peer review and any attached files.

Reviewer #1: **Yes: **Dr Matthew K Bagg

Reviewer #2: No

---

## [Author Response · Author response to Decision Letter 0]

14 Dec 2022

Dear editors and reviewers,

Thank you very much for your constructive comments and suggestions on our manuscript entitled “Comparative efficacy and acceptability of non-pharmacological interventions for depression among people living with HIV: A protocol for a systematic review and network meta-analysis” (PONE-D-22-17925). We appreciate the time and effort that you and the reviewers dedicated to providing feedback on our manuscript. We have carefully considered the comments and revised our manuscript accordingly. Changes to our previous manuscript are marked in red in the revised manuscript. The detailed point-by-point responses to the reviewer’s comments and concerns are listed below: 

Responses to the reviewers’ comments:

Reviewer #1:

Comment 1: You have defined the abbreviation for ART, yet not HAART. Can HAART be replaced with ART? I suggest rephrasing to "This systematic review protocol follows the Preferred Reporting Items for Systematic Review and Meta-analysis Protocols (PRISMA-P) 2015 statement. The study has been registered with PROSPERO (CRD42021244230)." 

Response 1: We sincerely thank you for your careful reading. We feel sorry for the typos in the original manuscript. We have removed the abbreviations to improve the readability of the manuscript because they appear less than three times in the text. We have also double-checked the English to polish the language throughout the manuscript. The revised texts read as follows: 

“As a life-altering disease associated with social stigma, comorbidities, adverse effects related to antiretroviral therapy, and uncertainty of prognosis, people living with HIV (PLWH) commonly experience psychological disorders[1-4]. The latest statistical analyses indicate that approximately 22%–44% of PLWH worldwide suffer from depression[5], which significantly contributes to poor HIV treatment outcomes, such as worse adherence to treatment, …” (Page 4, lines 53 to 58)

“This systematic review protocol follows the Preferred Reporting Items for Systematic Review and Meta-analysis Protocols (PRISMA-P) 2015 statement. The study has been registered with PROSPERO (CRD42021244230)” (Page 6, lines 100 to 102) 

Comment 2: *page 4, line 114*. Whilst often reasonable to exclude unpublished data, there is evidence in other fields that these data influence effect size. Is there any evidence in this field?

Response 2: Thanks for your valuable comments. We were previously concerned about the reliability of unpublished data as these do not undergo any peer review process, but we now realize that we need to provide unbiased estimates of treatment effects, so we have revised this aspect. We will include all eligible unpublished articles, the conference abstract, and the thesis. The electronic search will be supplemented with manual searches for published, unpublished, and ongoing randomized controlled trials in international trial registers (ClinicalTrials.gov, the International Clinical Trials Registry Platform (ICTRP) of the World Health Organization (WHO), and the EU Clinical Trials Register), ProQuest, and OpenGrey. We will also search for key conference proceedings in the field conducted by the American Psychiatric Association, the International College of Neuropsychology, the International AIDS Society, and the Conference on Retroviruses and Opportunistic Infections. Additional studies will be identified by hand-checking the bibliographies of the included studies and relevant review papers. (Page 9, lines 158 to 165)

Comment 3: *page 4, line 118*. It is not clear if the stated 'identification of mild depression based on standardized diagnostic criteria' is an inclusion criterion for the trials you will sample, or is a criterion you will apply to trials (based on their baseline charac.) yourself? I suspect the former, in which case it would read more clearly to say "We will include RCTs that sampled adults (age xxx) living with HIV and met standard diagnostic criteria for (at least) mild depression"

Response 3: We sincerely appreciate the valuable comments. Truly, we will include RCTs in which sampled adults living with HIV have met standardized diagnostic criteria for at least mild depression or have been identified as suffering from at least mild depression at baseline data based on any validated rating scale for depression. (Page 6, lines 111 to 114). 

Comment 4: *page 5, line 121*. I interpret your meaning by the statement 'For the clinical difference...' as indicating that you are excluding conditions that are clinically different to isolated depression. That being so, the wording is not clear and not ideal grammatically. Consider removing the 'clinical difference' statements in this paragraph. I think it then reads well for clarity and grammar.

Response 4: We sincerely thank you for your careful reading. We feel sorry for the wording in the original manuscript. We have polished the language to improve the readability of the manuscript. The revised texts read as follows: “We will exclude RCTs involving 20% or more participants with bipolar depression, treatment-resistant depression, or psychotic depression, but not studies involving participants with other comorbid psychiatric disorders (e.g., anxiety disorder).” (Pages 6 to 7, lines 114 to 117)

Comment 5: *page 5, lines 122-124*. This is not clear. Consider separating these sentences.

Response 5: Thank you for this valuable comment. The revised texts read as follows: “We will exclude RCTs involving 20% or more participants with bipolar depression, treatment-resistant depression, or psychotic depression, but not studies involving participants with other comorbid psychiatric disorders (e.g., anxiety disorder).” (Pages 6 to 7, lines 114 to 117)

Comment 6: *page 5, lines 133 to 134*. Firstly, please change 'category' to 'intervention' as this is clearer. Second, please justify excluding these trials? Are you not interested in the effect of dose? Either way, my experience is that often trials with 3 or more groups will test two doses of the same intervention against another intervention. You should not exclude these, because regardless of whether you combine the two doses into the same intervention node or not, the trial still contributes at least one comparison to the network.

Response 6: We sincerely appreciate the valuable comments. We apologize for the misunderstanding caused by our poor wording. In fact, we will consider different intensity levels or subtypes within the same type of non-pharmacological intervention as the same intervention node. We have polished the language to improve the readability of the manuscript. The revised texts read as follows: “We will consider comparative studies between different intensity levels or subtypes within the same type of non-pharmacological intervention as the same node in the network analysis.” (Page 7, lines 124 to 126)

Comment 7: *page 6, line 174*. Please expand these initials to include the first letter of all the names of each author?

Response 7: We sincerely thank you for your careful reading. The revised texts read as follows:

“… two investigators (TZ and HY) will independently screen all titles and abstracts identified in the searches.” (Page 9, lines 171 to 172)

“Two investigators (TZ and HY) will independently extract the data ...” (Page 10, line 181)

“Two investigators (TZ and HY) will independently assess the risk of bias in the included studies …” (Page 11, lines 205 to 206)

Comment 8: *page 7, line 195 to 196*. Grammar. Consider rephrasing to "When SDs are not reported...., and so on to calculate the SD.'

Response 8: Thanks for your valuable comments. We feel sorry for our poor wording in the original manuscript. We have double-checked and polished the grammar of the manuscript. The revised texts read as follows: “When SDs are not reported, we will first use the standard errors (SEs), t-statistics, p values, and so on to calculate the SDs.” (Pages 10 to 11, lines 197 to 198)

Comment 9: *page 7, line 198*. How many times?

Response 9: Thank you for this valuable comment. We will contact the authors of the included study up to three times via email to obtain the relevant missing data and seek clarifications, which will be regarded as unobtainable if no clarification is provided within six weeks. (Page 11, lines 198 to 201)

Comment 10: *page 7, line 201*. This is not ideal. I recommend at least including the study (if all other criteria are satisfied) and placing those lacking data in an Appendix. This helps gauge data availability biases in the field.

Response 10: Thanks for your valuable comments. We will place studies lacking data in the Appendix and explain why they were not included in the meta-analysis. The statement “We will exclude studies without relevant available data.” has also been removed from the manuscript to avoid potential ambiguity.

Comment 11: *page 7, line 203 to 204*. It is not clear what you will do here. How numerous are cluster trials in this field? It is common practice to exclude them from NMA.

Response 11: Thank you for this valuable comment. Cluster-randomized trials, which supplement traditional randomized clinical trials, are characterized by the randomization of groups—clusters—of patients[6]. The cluster could be health centers, hospitals, or villages. The Cochrane Handbook version 6.1.0 (Collaboration, 2020) [7] recommends that authors should always identify and examine any cluster-randomized trials in their reviews, as cluster-randomized trials usually have good plausibility. Therefore, we plan to include cluster-randomized clinical trials. However, it is noteworthy that in a cluster design, participants within any one cluster often tend to be homogeneous, so their data can no longer be assumed to be independent. In the context of a meta-analysis, studies in which clustering has been ignored will receive more weight than is appropriate. So, for cluster-randomized trials, we will extract data that account for the clustering in the results, such as the effect estimate adjusted for cluster correlation. (Pages 10 to 11, lines 195 to 196)

The following published articles of network meta-analysis reflected this point.

* Comparative efficacy and acceptability of first-generation and second-generation antidepressants in the acute treatment of major depression: protocol for a network meta-analysis. BMJ Open. 2016; 6(7): e010919. Doi: 10.1136/bmjopen-2015-010919.

* Comparative efficacy and acceptability of antidepressants in the long-term treatment of major depression: protocol for a systematic review and network meta-analysis. BMJ Open. 2019;9(5): e027574. Doi:10.1136/bmjopen-2018-027574.

Comment 12: *page 7, line 208*. 'assessed' is past tense - implying you have already done this. As this is a protocol, I suggest rephrasing to "Two investigators will independently assess methodological...."

Response 12: We sincerely thank you for your careful reading. We feel sorry for the typos in the original manuscript. We have double-checked and polished the grammar of the manuscript. The revised texts read as follows: “Two investigators (TZ and HY) will independently assess the risk of bias in the included studies …” (Page 11, lines 205 to 206)

Comment 13: *page 8, line 234*. The SMD has substantial limitations in the NMA context. There is the acknowledged problem of interpretability - what does proportion of unit deviation mean clinically? - that is a problem in pairwise meta-analysis as well. Specific to NMA, the validity of a SMD rests solely on the assumption of a common heterogeneity variance being clearly upheld. Have you thought about these issues? It is a good idea to use mean differences on a common scale, to which effects on different scales can be transformed. Do the clinimetric features of the outcome scales in this field permit this?

Response 13: Thank you for this valuable comment. We found two standardized metrics for the assessment of depression severity. However, these two metrics only include a limited number of depression symptom scales[8, 9]. Many depression scales could not be converted using these two metrics, such as the Hamilton Rating Scale of Depression, Montgomery Asberg Depression Rating Scale, Geriatric Depression Scale, Quick Inventory of Depression Symptoms-clinician Rating, Self-rating Depression Scale, Depression Anxiety and Stress Scale, and so on. If we use these two standardized metrics, we may need to exclude studies using the above scales, which would increase the risk of estimation bias. Furthermore, the validity of these two metrics has not been validated in external samples[10]. Therefore, we will not use these two standardized metrics to transform depression scores.

According to Cochrane Handbook version 6.1.0 (Collaboration, 2020)[7], the standardized mean difference (SMD) can be used as a summary statistic in a meta-analysis when the studies all assess the same outcome but measure it in a variety of ways. In this circumstance, it is necessary to standardize the results of the studies to a uniform scale before they can be combined[7]. Therefore, we decided to use the standardized mean differences in our network meta-analysis. However, given the importance of the heterogeneity assumption in network meta-analysis, we will assume a common heterogeneity variance parameter for each network and present the estimate for this parameter from the network meta-analysis model. If significant heterogeneity is found, we will consider the random-effects model in our analysis and conduct subgroup analyses or network meta-regression analyses to explore the possible sources. (Page 14, lines 266 to 270; Pages 15 to 16, lines 296 to 310)

The following published articles of network meta-analysis also used SMD to synthesize data on depression scores from different tools. 

* A network meta-analysis of the effects of psychotherapies, pharmacotherapies and their combination in the treatment of adult depression. World Psychiatry. 2020;19(1):92-107. doi:10.1002/wps.20701

* Comparative efficacy and acceptability of antidepressants, psychological interventions, and their combination for depressive disorder in children and adolescents: protocol for a network meta-analysis. BMJ Open. 2017;7(8):e016608. Published 2017 Aug 11. doi:10.1136/bmjopen-2017-016608

* Comparative efficacy of non-pharmacological interventions on agitation in people with dementia: A systematic review and Bayesian network meta-analysis. Int J Nurs Stud. 2020;102:103489. doi:10.1016/j.ijnurstu.2019.103489

Comment 14: *page 8, lines 237 to 239*. A few clarifications. Heterogeneity is correctly estimated by estimating tau^2 (aka the heterogeneity variance aka the variance of the random effects distribution). Cochran's Q is not an estimate, rather it is a chi^2 test of difference between study observations and the estimated mean of the random effects distribution (the pooled effect point estimate). I^2 is not a statistic, rather it is a proportional measure of variability. A larger I^2 indicates that a larger relative proportion of the observed heterogeneity is due to true variability between studies (true heterogeneity) rather than sampling error. See Borenstein et al. *Res Synth Meth*. DOI: 10.1002/jrsm.1230. Accordingly, I recommend rephrasing these lines and replacing 'estimated' with 'evaluated', as this term encompasses estimation, testing and measures.

Response 14: Many thanks to your kind reminder. The revised texts read as follows: “Heterogeneity will be evaluated using Cochran’s Q test and I2 test. P<0.10 means the heterogeneity is statistically significant. I2 reflects the level of heterogeneity. If I2 >50%, which indicates substantial heterogeneity, we will consider subgroup analysis to explore any identified heterogeneity.” (Page 13, lines 240 to 243)

Comment 15: *page 8, line 240*. Please add '...subgroup analysis to explore any identified heterogeneity' for clarity?

Response 15: Many thanks to your kind reminder. The revised texts read as follows: 

“If I2 >50%, which indicates substantial heterogeneity, we will consider subgroup analysis to explore any identified heterogeneity.” (Page 13, lines 241 to 243)

Comment 16: *page 9, line 248, 266 to 269*. Netmeta implements a quasi-frequentist approach, not Bayesian. There are several R packages for Bayesian NMA. Please clarify which packages you will use? As well, SUCRA scores are not calculable with netmeta - rather P-scores, which are somewhat analogous. See Rucker & Schwarzer 2015 *BMC MRM*. DOI: 10.1186/s12874-015-0060-8.

Response 16: Thank you for this valuable comment. We have carefully read the description of the ‘netmeta’ package and found that this package is a set of functions providing frequentist methods for network meta-analysis[11]. After an extensive literature review, we decided to use OpenBUGS (version 3.2.3) program to fit our model. OpenBUGS is an open source software for Bayesian statistics using Markov Chain Monte Carlo simulation, which is a highly versatile tool for applying Bayesian methodology[12, 13]. What’s more, the importance of the software has also been widely acknowledged[14]. The revised texts read as follows: 

“We will fit our models using OpenBUGS (version 3.2.3) software. … We will set the OpenBUGS program to run three Markov chains simultaneously with different arbitrarily chosen initial values.” (Page 13, lines 259 to 262)

Comment 17: *page 9, line 248 to 251*. What priors will you use for the analysis of efficacy? Is 'no prior' different to 'uninformative'?

Response 17: Many thanks to your kind reminder. We feel sorry for our careless mistakes. Bayesian methods involve a formal combination of a prior probability distribution to obtain a corresponding posterior probability distribution. The posterior distribution, as obtained with the Bayesian approach, can be interpreted in terms of probabilities (e.g., “There is an x% probability that treatment A results in a greater response than treatment B”). To not influence the observed results by the prior distribution, an often-heard critique of the Bayesian approach, we used a noninformative prior distribution for the treatment effect parameters (efficacy and acceptability). With such a “flat” prior distribution, it is assumed that before seeing the data, any parameter value is “equally” likely. Therefore, posterior results are not influenced by the prior distribution but are driven by the data[15]. 

Actually, 'No prior' is equal to 'uninformative'. We have modified the wording in our revised manuscript to avoid potential ambiguity. The revised texts read as follows: “Uninformative prior distributions will be used for all model parameters.” (Page 13, lines 256 to 257)

Comment 18: *page 9, line 255 to 261*. This is not good practice. You should make this decision *a priori* based on clinical judgement. An important consideration that is not mentioned is whether you will fit a coherence or incoherence model? 

Response 18: We sincerely appreciate the valuable comments. We reviewed a large body of literature and found that your point is correct. Especially in Network Meta-Analysis for Decision Making, Dias et al. stated, “Model choice should be based on contextual factors as well as on statistical measures of model fit and adequacy. For example, if inclusion criteria are strict, then fixed effects models are plausible; however, where populations and interventions are heterogeneous, we may not consider a fixed effects model plausible, even if model fit statistics indicate it to be the most parsimonious model. This is especially the case in situations where the evidence available to inform a network meta‐analysis is sparse, so that there is very little power to detect heterogeneity if it exists.” Therefore, although we have strict inclusion criteria, we will still consider both fixed-effects models and random-effects models in the Bayesian analyses due to the anticipated heterogeneity. In addition, we will also calculate the posterior mean of residual deviance to access model fit. The closer the posterior mean of residual deviance is to the number of data points, the better the model fit[16]. (Page 13, lines 251 to 255)

We will choose the consistency or inconsistency model based on the results of the inconsistency tests. Specifically, we perform inconsistency tests through the side-splitting approach[17], the loop-specific approach[18], and the design-by-treatment test[19]. If there is no obvious inconsistency, we will use the consistency model; otherwise, an inconsistency model will be used[17, 19]. (Page 15, lines 285 to 291)

Comment 19: *page 9, line 262*. It's convention to describe this as the 'common heterogeneity variance parameter'.

Response 19: Many thanks to your kind reminder. The revised texts read as follows: “We will assume a common heterogeneity variance parameter for each network and present the estimate for this parameter from the network meta-analysis model. We will also estimate the prediction intervals to evaluate to what degree the common heterogeneity affects the relative effects of the additional uncertainty anticipated in future studies.” (Page 14, lines 266 to 270)

Comment 20: *page 10, line 283*. The methods available for coherence assessment will depend on the package.

Response 20: Thank you for this valuable comment. Our all analyses for inconsistency will be conducted using the ‘network’ and ‘mvmeta’ packages in Stata (version 15.1) software. Specifically, we will use the design-by-treatment test[19] to assess inconsistency in the entire network. Side-splitting analysis[17] and the loop-specific method[18] will be employed to evaluate local inconsistency. (Page 15, lines 285 to 293)

Comment 21: *page 10, line 296 to 297*. These models need further description. Why will you only consider sample size? Will you fit NMReg models to explore identified heterogeneity/incoherence, or regardless? What are your assumptions about the regression coefficients? See Donegan et al. *Res Synth Meth*. DOI: 10.1002/jrsm.1327

Response 21: We sincerely appreciate the valuable comments. We apologize for the ambiguity caused by our poor wording. In fact, we aim to employ network meta-regression to account for small-study effects. Given the potential ambiguity, we decided to use the comparison-adjusted funnel plot to assess the presence of small-study effects and publication bias in the network. Furthermore, Egger’s test will be used to examine the significant publication bias. (Page 16, lines 318 to 320)

If we find significant heterogeneity and/or inconsistency, we will conduct subgroup analyses or network meta-regression analyses to explore the possible sources using the following characteristics: (1) sample size; (2) the severity of depressive symptoms at baseline; (3) the treatment duration; (4) the risk of bias. We assume consistent relative treatment effects estimated at the covariate value 0 or at the mean covariate value and consistent regression coefficients for the treatment by covariate interaction[20]. We hypothesize that: 

Increasing sample size reduces the effect size between intervention and comparator.

Increasing baseline depression severity increases the effect size between intervention and comparator.

Increasing treatment duration increases the effect size between intervention and comparator.

Increasing risk of bias increases the effect size between intervention and comparator. (Pages 15 to 16, lines 296 to 310) 

Comment 22: *page 10, line 311*. Tense. 'Use' is present/future tense.

Response 22: We sincerely thank you for your careful reading. We feel sorry for the typos in the original manuscript. We have double-checked and polished the grammar of the manuscript. The revised texts read as follows: “We will use the Confidence In Network Meta­Analysis (CINeMA) software (a web application based on the GRADE framework) during the evaluation[61, 62].” (Page 17, lines 329 to 331)

Special thanks to you for your valuable comments.

Reviewer #2:

 Comment 1: Abstract. “We will employ a network meta-analysis to synthesize all available evidence for each outcome and obtain a comprehensive ranking of all interventions for the global network and for low-income and middle-income countries network only.” I’m not sure what network is being referenced. Also, should it be countries’ network? 

Response 1: Thank you for this valuable comment. We apologize for the ambiguity caused by our poor wording. We polished the language to improve the readability of the manuscript. The revised texts read as follows: “We will employ a network meta-analysis to synthesize all available evidence for each outcome and obtain a comprehensive ranking of all treatments for the global network of countries as well as for the network of LMICs only.” (Page 3, lines 40 to 43)

Comment 2: Paragraph 2, sentence 3 “Furthermore, it may cause potential somatic distress, drug interactions with HIV medications, drug abuse, and lethality in overdose [13].” The support is poor as it is from the introduction to a 2009 paper and not primary literature. The above sentence in the paper needs better citation given pharmacotherapy is standard of care in the general population in high income countries and generally safe. I’m not tracking with how antidepressants are causing drug abuse. Also, most SSRIs are hard to overdose on.

Response 2: We sincerely appreciate the valuable comments. We have updated the citations to update and support this point. For drug abuse, evidence suggests that while ketamine has a significant and rapid antidepressant effect, it also has a high potential of drug abuse[23, 24]. For lethality in overdose, European AIDS Clinical Society guidelines indicate that selective serotonin-reuptake inhibitors (SSRIs, e.g., paroxetine, sertraline, and citalopram), mixed or dual-action reuptake inhibitors (e.g., venlafaxine), and mixed-action newer agents have a moderate or low risk of the lethality of overdose[25]. The revised texts read as follows: “… Although pharmacotherapy is a cornerstone of depression management in PLWH, there are concerns about certain adverse effects (e.g., potential somatic distress, sexual dysfunction, drug abuse, and lethality in overdose), drug interactions with HIV medications, and withdrawal and rebound phenomena[23-28]. This has caused non-pharmacological treatments for depression in PLWH to become increasingly popular.” (Page 4, lines 61 to 66)

Comment 3: Paragraph 3- low cost- varies widely. Depends on who is giving it. You go on to explain why it is not low cost- requirement of trained staff and the limited access to such staff.

Response 3: Thanks for your valuable comments. After reviewing the literature, we found that psychotherapy does have relatively high costs, including direct treatment costs, additional direct costs (auxiliary psychiatric treatments during the follow-up period), and indirect costs, given the limited mental health resources available[29-31]. Therefore, we have removed the sentence from the revised manuscript. The revised texts read as follows: “There is a wide choice of non-pharmacological treatments, including psychotherapy (e.g., cognitive behavioral therapy)[32, 33], psychosocial interventions (e.g., meditation and yoga)[34-36], physical therapies (e.g., exercise)[37-40], music therapy[41], and so on. Among them, clinical practice guidelines recommend psychotherapy as the first-line treatment for depression[42, 43]. However, such interventions rely on mental health professionals for delivery. In most countries, particularly low-income and middle-income countries (LMICs), the limited availability and accessibility of psychotherapy resources restrict the application of mental health services[44, 45]. Evidence suggests that the rate of available mental health therapists per 100,000 people in LMICs is about 0.5% of that in high-income countries[45-47], and in some cases, only one or two psychiatrists are available for the entire country[46]. Thus, some treatments that can complement or alternate with psychotherapy remain indispensable …” (Pages 4 to 5, lines 67 to 79)

Comment 4: Paragraph 4- You are implying these non-pharmacologic interventions bring benefits. I think you are actually weighing if they bring benefit and how much.

Response 4: Thank you for this valuable comment. We agree with this point. We aim to identify the most effective and acceptable treatment among all available non-pharmacological interventions for depression in people living with HIV (PLWH). The revised texts read as follows: “… although many non-pharmacological resources have been shown to effectively alleviate depression in PLWH, these non-pharmacological treatments are rarely systematically compared. Traditional pairwise meta-analyses could investigate only the relative effect of a single direct comparison (usually a treatment compared to the usual care or waiting lists), rather than performing multiple comparisons simultaneously, and this is further limited by the lack of availability of a sufficient number of head-to-head comparisons between different treatments[48]. The most effective and acceptable non-pharmacological treatment for depression among PLWH in different resource contexts is yet to be identified by further research.” (Page 5, lines 79 to 88) 

Comment 5: Grammatical errors- ex. Like 87-88 should be “have been provided.” Not “has been provided”.

Response 5: We sincerely thank you for your careful reading. We feel sorry for the grammatical errors in the original manuscript. We have double-checked and polished the grammar of the manuscript. The revised texts read as follows: “… although many non-pharmacological resources have been shown to effectively alleviate depression in PLWH, …” (Page 5, lines 79 to 80)

Comment 6: Network meta-analysis- should have a citation.

Response 6: Many thanks to your kind reminder. The citation has been added (Page 5, lines 89 to 91).

Comment 7: Types of Participants. “For the clinical difference, we will exclude RCTs in which 20% or more of the participants are suffering from bipolar or psychotic depression, but not involving patients with other comorbid psychiatric disorders (eg. anxiety disorder).” I am not sure what this is saying. Please re-word the sentence.

Response 7: Thank you for this valuable comment. We have polished the language to express more accurately in the revised manuscript. The revised texts read as follows: “We will exclude RCTs involving 20% or more participants with bipolar depression, treatment-resistant depression, or psychotic depression, but not studies involving participants with other comorbid psychiatric disorders (e.g., anxiety disorder).” (Pages 6 to 7, lines 114 to 117) 

Comment 8: “We will also exclude studies in which participants with a serious concomitant medical illness. “HIV is a serious medical illness. I think you mean other than HIV but the sentence should be reworded.

Response 8: We sincerely appreciate the valuable comments. The revised texts read as follows: “We will also exclude studies in which participants have serious concomitant medical illnesses other than HIV.” (Page 7, lines 118 to 119)

Comment 9: Types of Intervention. Synthetic comparison set- needs a definition.

Response 9: Thank you for this valuable comment. We apologize for the ambiguity caused by our poor wording. We have modified the wording in our revised manuscript. The revised texts read as follows: “All the eligible interventions will constitute a network of interventions. In principle, all eligible participants are equally likely to be randomly assigned to any interventions in the network.” (Page 7, lines 126 to 128)

Comment 10: Table 1 has inconsistent capitalization.

Response 10: Many thanks to your kind reminder. We apologize for the typos in the original manuscript. We have standardized the capitalization in Table 1 of the revised manuscript (Page 8, Table 1). We have also double-checked the English to polish the language throughout the manuscript.

Special thanks to you for your valuable comments.

We are very grateful to you and the reviewers for your valuable comments on our paper. If there are any other modifications we could make, we would like to modify them further, and we appreciate your help. We hope that our manuscript can be considered for publication in your journal. Looking forward to hearing from you.

Thank you and best regards.

Yours sincerely,

Honghong Wang

Xiangya School of Nursing, Central South University, 172 Tongzuphill, Yuelu District, Changsha, Hunan Province, P. R. China, 410013.

Email: honghong_wang@hotmail.com

Reference

1. Cheng LJ, Kumar PA, Wong SN, Lau Y. Technology-delivered psychotherapeutic interventions in improving depressive symptoms among people with HIV/AIDS: a systematic review and meta-analysis of randomised controlled trials. AIDS Behav. 2020;24(6):1663-75. Epub 2019/10/07. doi: 10.1007/s10461-019-02691-6. PubMed PMID: 31587115.

2. Lazarus J, Safreed-Harmon K, Barton S, Costagliola D, Dedes N, Del Amo Valero J, et al. Beyond viral suppression of HIV - the new quality of life frontier. BMC medicine. 2016;14(1):94. doi: 10.1186/s12916-016-0640-4. PubMed PMID: 27334606.

3. Nanni MG, Caruso R, Mitchell AJ, Meggiolaro E, Grassi L. Depression in HIV infected patients: a review. Curr Psychiatry Rep. 2015;17(1):530. Epub 2014/11/22. doi: 10.1007/s11920-014-0530-4. PubMed PMID: 25413636.

4. van der Heijden I, Abrahams N, Sinclair D. Psychosocial group interventions to improve psychological well-being in adults living with HIV. Cochrane Database Syst Rev. 2017;3(3):Cd010806. Epub 2017/03/16. doi: 10.1002/14651858.CD010806.pub2. PubMed PMID: 28291302.

5. Rezaei S, Ahmadi S, Rahmati J, Hosseinifard H, Dehnad A, Aryankhesal A, et al. Global prevalence of depression in HIV/AIDS: a systematic review and meta-analysis. BMJ Support Palliat Care. 2019;9(4):404-12. Epub 2019/09/21. doi: 10.1136/bmjspcare-2019-001952. PubMed PMID: 31537580.

6. Fayers PM, Jordhøy MS, Kaasa S. Cluster-randomized trials. Palliat Med. 2002;16(1):69-70. Epub 2002/04/20. doi: 10.1191/0269216302pm503xx. PubMed PMID: 11963457.

7. Higgins JPT TJ, Chandler J, Cumpston M, Li T, Page MJ, Welch VA (editors). Cochrane handbook for systematic reviews of interventions version 6.1 (updated September 2020). Cochrane2020. Available from: www.training.cochrane.org/handbook.

8. Choi SW, Schalet B, Cook KF, Cella D. Establishing a common metric for depressive symptoms: linking the BDI-II, CES-D, and PHQ-9 to PROMIS depression. Psychol Assess. 2014;26(2):513-27. Epub 2014/02/20. doi: 10.1037/a0035768. PubMed PMID: 24548149; PubMed Central PMCID: PMCPMC5515387.

9. Wahl I, Löwe B, Bjorner JB, Fischer F, Langs G, Voderholzer U, et al. Standardization of depression measurement: a common metric was developed for 11 self-report depression measures. J Clin Epidemiol. 2014;67(1):73-86. Epub 2013/11/23. doi: 10.1016/j.jclinepi.2013.04.019. PubMed PMID: 24262771.

10. Fischer HF, Rose M. www.common-metrics.org: a web application to estimate scores from different patient-reported outcome measures on a common scale. BMC Med Res Methodol. 2016;16(1):142. Epub 2016/10/21. doi: 10.1186/s12874-016-0241-0. PubMed PMID: 27760525.

11. Gerta Rücker UK, Jochem König, Orestis Efthimiou, Annabel Davies, Theodoros Papakonstantinou, Guido Schwarzer. netmeta: Network Meta-Analysis using Frequentist Methods 2022. Available from: https://cran.r-project.org/web/packages/netmeta/index.html.

12. David Spiegelhalter AT, Nicky Best, Dave Lunn. OpenBUGS User Manual 2014. Available from: https://www.mrc-bsu.cam.ac.uk/wp-content/uploads/2021/06/OpenBUGS_Manual.pdf.

13. Congdon P. Bayesian statistical modelling: John Wiley & Sons; 2007.

14. Lunn D, Spiegelhalter D, Thomas A, Best N. The BUGS project: Evolution, critique and future directions. Stat Med. 2009;28(25):3049-67. Epub 2009/07/25. doi: 10.1002/sim.3680. PubMed PMID: 19630097.

15. Hoaglin DC, Hawkins N, Jansen JP, Scott DA, Itzler R, Cappelleri JC, et al. Conducting indirect-treatment-comparison and network-meta-analysis studies: report of the ISPOR task force on indirect treatment comparisons good research practices: part 2. Value Health. 2011;14(4):429-37. Epub 2011/06/15. doi: 10.1016/j.jval.2011.01.011. PubMed PMID: 21669367.

16. Dias S, Ades AE, Welton NJ, Jansen JP, Sutton AJ. Network meta-analysis for decision-making: John Wiley & Sons; 2018.

17. Dias S, Welton NJ, Caldwell DM, Ades AE. Checking consistency in mixed treatment comparison meta-analysis. Stat Med. 2010;29(7-8):932-44. Epub 2010/03/10. doi: 10.1002/sim.3767. PubMed PMID: 20213715.

18. Bucher HC, Guyatt GH, Griffith LE, Walter SD. The results of direct and indirect treatment comparisons in meta-analysis of randomized controlled trials. Journal of Clinical Epidemiology. 1997;50(6):683-91. doi: 10.1016/S0895-4356(97)00049-8.

19. Higgins JP, Jackson D, Barrett JK, Lu G, Ades AE, White IR. Consistency and inconsistency in network meta-analysis: concepts and models for multi-arm studies. Res Synth Methods. 2012;3(2):98-110. Epub 2012/06/01. doi: 10.1002/jrsm.1044. PubMed PMID: 26062084.

20. Donegan S, Dias S, Welton NJ. Assessing the consistency assumptions underlying network meta-regression using aggregate data. Res Synth Methods. 2019;10(2):207-24. Epub 2018/10/28. doi: 10.1002/jrsm.1327. PubMed PMID: 30367548.

21. Puhan MA, Schünemann HJ, Murad MH, Li T, Brignardello-Petersen R, Singh JA, et al. A GRADE working group approach for rating the quality of treatment effect estimates from network meta-analysis. BMJ. 2014;349:g5630. Epub 2014/09/26. doi: 10.1136/bmj.g5630. PubMed PMID: 25252733.

22. Nikolakopoulou A, Higgins JPT, Papakonstantinou T, Chaimani A, Del Giovane C, Egger M, et al. CINeMA: an approach for assessing confidence in the results of a network meta-analysis. PLoS Med. 2020;17(4):e1003082. Epub 2020/04/04. doi: 10.1371/journal.pmed.1003082. PubMed PMID: 32243458.

23. Liu Y, Lin D, Wu B, Zhou W. Ketamine abuse potential and use disorder. Brain Res Bull. 2016;126(Pt 1):68-73. Epub 2016/06/05. doi: 10.1016/j.brainresbull.2016.05.016. PubMed PMID: 27261367.

24. Trujillo KA, Iñiguez SD. Ketamine beyond anesthesia: antidepressant effects and abuse potential. Behav Brain Res. 2020;394:112841. Epub 2020/08/03. doi: 10.1016/j.bbr.2020.112841. PubMed PMID: 32739287.

25. Society EAC. European AIDS clinical society guidelines 2022 [cited 2022 October]. Available from: https://www.eacsociety.org/media/guidelines-11.1_final_09-10.pdf.

26. Ferrando S. Psychopharmacologic treatment of patients with HIV/AIDS. Current psychiatry reports. 2009;11(3):235-42. doi: 10.1007/s11920-009-0036-7. PubMed PMID: 19470286.

27. Henssler J, Heinz A, Brandt L, Bschor T. Antidepressant withdrawal and rebound phenomena. Dtsch Arztebl Int. 2019;116(20):355-61. Epub 2019/07/11. doi: 10.3238/arztebl.2019.0355. PubMed PMID: 31288917.

28. Fava GA, Cosci F. Understanding and managing withdrawal syndromes after discontinuation of antidepressant drugs. J Clin Psychiatry. 2019;80(6). Epub 2019/11/28. doi: 10.4088/JCP.19com12794. PubMed PMID: 31774947.

29. König H, König HH, Konnopka A. The excess costs of depression: a systematic review and meta-analysis. Epidemiol Psychiatr Sci. 2019;29:e30. Epub 2019/04/06. doi: 10.1017/s2045796019000180. PubMed PMID: 30947759.

30. Maljanen T, Knekt P, Lindfors O, Virtala E, Tillman P, Härkänen T. The cost-effectiveness of short-term and long-term psychotherapy in the treatment of depressive and anxiety disorders during a 5-year follow-up. J Affect Disord. 2016;190:254-63. Epub 2015/11/06. doi: 10.1016/j.jad.2015.09.065. PubMed PMID: 26540079.

31. Watkins KE, Burnam MA, Orlando M, Escarce JJ, Huskamp HA, Goldman HH. The health value and cost of care for major depression. Value Health. 2009;12(1):65-72. Epub 2009/11/17. doi: 10.1111/j.1524-4733.2008.00388.x. PubMed PMID: 19911440.

32. Spies G, Asmal L, Seedat S. Cognitive-behavioural interventions for mood and anxiety disorders in HIV: a systematic review. J Affect Disord. 2013;150(2):171-80. Epub 2013/05/22. doi: 10.1016/j.jad.2013.04.018. PubMed PMID: 23688915.

33. Crepaz N, Passin WF, Herbst JH, Rama SM, Malow RM, Purcell DW, et al. Meta-analysis of cognitive-behavioral interventions on HIV-positive persons' mental health and immune functioning. Health Psychol. 2008;27(1):4-14. Epub 2008/01/31. doi: 10.1037/0278-6133.27.1.4. PubMed PMID: 18230008.

34. Jiang T, Hou J, Sun R, Dai L, Wang W, Wu H, et al. Immunological and psychological efficacy of meditation/yoga intervention among people living with HIV (PLWH): a systematic review and meta-analyses of 19 randomized controlled trials. Ann Behav Med. 2020;55(6):505-19. Epub 2020/10/30. doi: 10.1093/abm/kaaa084. PubMed PMID: 33119732.

35. Scott-Sheldon LAJ, Balletto BL, Donahue ML, Feulner MM, Cruess DG, Salmoirago-Blotcher E, et al. Mindfulness-based interventions for adults living with HIV/AIDS: a systematic review and meta-analysis. AIDS Behav. 2019;23(1):60-75. Epub 2018/07/29. doi: 10.1007/s10461-018-2236-9. PubMed PMID: 30054765.

36. Kuloor A, Kumari S, Metri K. Impact of yoga on psychopathologies and quality of life in persons with HIV: a randomized controlled study. J Bodyw Mov Ther. 2019;23(2):278-83. Epub 2019/05/20. doi: 10.1016/j.jbmt.2018.10.005. PubMed PMID: 31103108.

37. O'Brien K, Nixon S, Tynan AM, Glazier R. Aerobic exercise interventions for adults living with HIV/AIDS. Cochrane Database Syst Rev. 2010;2010(8):Cd001796. Epub 2010/08/06. doi: 10.1002/14651858.CD001796.pub3. PubMed PMID: 20687068.

38. Heissel A, Zech P, Rapp MA, Schuch FB, Lawrence JB, Kangas M, et al. Effects of exercise on depression and anxiety in persons living with HIV: a meta-analysis. J Psychosom Res. 2019;126:109823. Epub 2019/09/14. doi: 10.1016/j.jpsychores.2019.109823. PubMed PMID: 31518734.

39. Reychler G, Caty G, Arcq A, Lebrun L, Belkhir L, Yombi JC, et al. Effects of massage therapy on anxiety, depression, hyperventilation and quality of life in HIV infected patients: a randomized controlled trial. Complement Ther Med. 2017;32:109-14. Epub 2017/06/18. doi: 10.1016/j.ctim.2017.05.002. PubMed PMID: 28619295.

40. Oliveira VHF, Rosa FT, Santos JC, Wiechmann SL, Narciso AMS, Franzoi de Moraes SM, et al. Effects of a combined exercise training program on health indicators and quality of life of people living with HIV: a randomized clinical trial. AIDS Behav. 2020;24(5):1531-41. Epub 2019/09/26. doi: 10.1007/s10461-019-02678-3. PubMed PMID: 31552510.

41. Aalbers S, Fusar-Poli L, Freeman RE, Spreen M, Ket JC, Vink AC, et al. Music therapy for depression. Cochrane Database Syst Rev. 2017;11(11):Cd004517. Epub 2017/11/17. doi: 10.1002/14651858.CD004517.pub3. PubMed PMID: 29144545.

42. Organization WH. mhGAP intervention guide. Geneva: World Health Organization; 2010.

43. Cleare A, Pariante CM, Young AH, Anderson IM, Christmas D, Cowen PJ, et al. Evidence-based guidelines for treating depressive disorders with antidepressants: a revision of the 2008 British Association for psychopharmacology guidelines. J Psychopharmacol. 2015;29(5):459-525. Epub 2015/05/15. doi: 10.1177/0269881115581093. PubMed PMID: 25969470.

44. Domhardt M, Baumeister H. Psychotherapy of adjustment disorders: current state and future directions. World J Biol Psychiatry. 2018;19(sup1):S21-s35. Epub 2018/09/12. doi: 10.1080/15622975.2018.1467041. PubMed PMID: 30204563.

45. Saxena S, Thornicroft G, Knapp M, Whiteford H. Resources for mental health: scarcity, inequity, and inefficiency. Lancet (London, England). 2007;370(9590):878-89. doi: 10.1016/s0140-6736(07)61239-2. PubMed PMID: 17804062.

46. Naslund J, Aschbrenner K, Araya R, Marsch L, Unützer J, Patel V, et al. Digital technology for treating and preventing mental disorders in low-income and middle-income countries: a narrative review of the literature. The lancet Psychiatry. 2017;4(6):486-500. doi: 10.1016/s2215-0366(17)30096-2. PubMed PMID: 28433615.

47. Chisholm D, Sweeny K, Sheehan P, Rasmussen B, Smit F, Cuijpers P, et al. Scaling-up treatment of depression and anxiety: a global return on investment analysis. Lancet Psychiatry. 2016;3(5):415-24. Epub 2016/04/17. doi: 10.1016/s2215-0366(16)30024-4. PubMed PMID: 27083119.

48. Lu G, Ades AE. Combination of direct and indirect evidence in mixed treatment comparisons. Stat Med. 2004;23(20):3105-24. Epub 2004/09/28. doi: 10.1002/sim.1875. PubMed PMID: 15449338.

---

## [Decision Letter · Decision Letter 1]

20 Apr 2023

PONE-D-22-17925R1Comparative efficacy and acceptability of non-pharmacological interventions for depression among people living with HIV: A protocol for a systematic review and network meta-analysisPLOS ONE

Dear Dr. Wang,

Thank you for submitting your manuscript to PLOS ONE. After careful consideration, we feel that it has merit but does not fully meet PLOS ONE’s publication criteria as it currently stands. Therefore, we invite you to submit a revised version of the manuscript that addresses the points raised during the review process.

We look forward to receiving your revised manuscript.

Kind regards,

Guglielmo Campus, Ph.D DDS

Academic Editor

PLOS ONE

Journal Requirements:

Reviewers' comments:

Reviewer's Responses to Questions

**Comments to the Author**

1. Does the manuscript provide a valid rationale for the proposed study, with clearly identified and justified research questions?

Reviewer #1: Yes

Reviewer #3: Yes

2. Is the protocol technically sound and planned in a manner that will lead to a meaningful outcome and allow testing the stated hypotheses?

Reviewer #1: Yes

Reviewer #3: Yes

3. Is the methodology feasible and described in sufficient detail to allow the work to be replicable?

Reviewer #1: Yes

Reviewer #3: Yes

4. Have the authors described where all data underlying the findings will be made available when the study is complete?

Reviewer #1: No

Reviewer #3: Yes

5. Is the manuscript presented in an intelligible fashion and written in standard English?

Reviewer #1: Yes

Reviewer #3: Yes

6. Review Comments to the Author

You may also provide optional suggestions and comments to authors that they might find helpful in planning their study.

Reviewer #1: Thank you for your responses and the changes you have made. I am pleased we have been able to work together effectively.

I would like to make some follow-up points.

Regarding unpublished data, you are correct in holding concerns about the veracity of these data. It is also an issue that we can often not appraise ROB adequately from a trial registry report alone. I suggest you factor this into your assessment of these trials. I am pleased overall that you will collect them. If there is no evidence yet as to their effect on overall meta-analytic effect sizes, perhaps you will be in a position to demonstrate that for the field.

Thank you for clarifying the exclusion of studies with mixed samples. I suggest an additional sentence to clarify the reason for exclusion of other types of depression being that you think these will interact with the estimated effects (?), which is not the case for co-morbid psychiatric disorders. In effect, you are conditioning by design, although that won't be clear to everyone.

Thank you for clarifying your node definitions. I would keep an open mind to the possibility of splitting nodes according to levels of dose, because you may find adequate data with which to do this (more commonly we don't). As you have it written, defining multiple nodes for different doses of the same intervention would be a protocol deviation, which would be unfortunate. I suggest "We will consider different intensity levels or subtypes within the same type of non-pharmacological intervention as the same node in the network analysis in the first instance, or as different nodes if adequate data are available.”

Great. That's entirely appropriate to handle clustered trials in that manner.

Okay, your reasons for using SMD are considered and reasonable. In that case, please bear in mind that the estimate of the heterogeneity variance will be critical to accurate interpretation of the results. You are assuming this parameter to be common across comparisons - that's reasonable and often the only option given the amount of data. That being so, if this estimate is not precise, it will be difficult to interpret the SMDs for the relative effects.

Thanks for making those updates regarding heterogeneity estimation and exploration. Strictly, I^2 is not a test, so please drop that word.

Fair enough regarding priors. Bear in mind that an informative prior can be justified. In fact for binary outcomes, and some continuous outcomes, there are extant empirical distributions for selecting priors (10.1002/sim.9076; 10.1007/s11606-020-06357-1; 10.1002/jrsm.1475; 10.1093/ije/dys041; 10.1016/j.jclinepi.2014.08.012).

Thank you, I agree with that approach regarding choice of a consistency vs inconsistency model, and your investigation of coherence. You might also note that you can compare the fit of the two models as an additional 'global' index of incoherence (although null difference is not evidence of coherence).

I urge you strongly not to compare results of models with fixed or random effects terms for the treatment effects. Whereas with coherence we often do not know, in the case of whether treatment effects follow a distribution (random setting) or vary only due to sampling error (fixed setting), it is rare this is not known. Please choose an approach and stick with it, because the choice of term influences the ability to assess coherence. For example, in a network where there are few studies per comparison, and the model has a random effects term for treatment effects, there may be a false conclusion of no loop incoherence simply because the CIs are enlarged (and thus overlap for the indirect and direct estimates). Whilst, in this hypothetical scenario, the random effects term may be appropriate, there is the noted drawback of lesser ability to evaluate coherence. You won't have major problems, because you're using multiple means to evaluate coherence, but the point still stands that the treatment effects term needs to be chosen based on prior knowledge to avoid unwarranted model flexibility.

Please bear in mind the potential for results to vary between the Bayesian and frequentist settings. I suggest you also estimate treatment effects with mvmeta, compare them to those from OpenBUGS, and note the differences before looking at coherence metrics.

Lastly, please add a statement of where you will disseminate the study findings and data?

Overall, this revised manuscript was a pleasure to read. Good luck with the study

Reviewer #3: I would like to compliment the authors for their revision. However, it would be grateful if they included a sentence in the methods on how to deal with missing data and what procedure to consider in the case of incomparable nodes of the interventions examined. I also believe that the overall quality of evidence for each treatment comparison (on the main outcomes) should be done using the Confidence in Network Meta-Analysis (CINeMA) tool (https://cinema.ispm.unibe.ch).

7. PLOS authors have the option to publish the peer review history of their article (what does this mean?). If published, this will include your full peer review and any attached files.

Reviewer #1: **Yes: **Matthew K Bagg

Reviewer #3: No

---

## [Author Response · Author response to Decision Letter 1]

17 May 2023

Dear editors and reviewers,

Thank you very much for your constructive comments and suggestions on our manuscript entitled “Comparative efficacy and acceptability of non-pharmacological interventions for depression among people living with HIV: A protocol for a systematic review and network meta-analysis” (PONE-D-22-17925). We appreciate the time and effort that you and the reviewers dedicated to providing feedback on our manuscript. We have carefully considered the comments and revised our manuscript accordingly. Changes to our previous manuscript are marked in red on the revised manuscript. The detailed point-by-point responses to the reviewer’s comments and concerns are listed below: 

Responses to Journal Requirements:

 Comment 1: Please review your reference list to ensure that it is complete and correct. If you have cited papers that have been retracted, please include the rationale for doing so in the manuscript text, or remove these references and replace them with relevant current references. Any changes to the reference list should be mentioned in the rebuttal letter that accompanies your revised manuscript. If you need to cite a retracted article, indicate the article’s retracted status in the References list and also include a citation and full reference for the retraction notice.

 Response 1: We sincerely appreciate the valuable comments. We have ensured that our reference list is complete and correct. We have not cited papers that have been retracted. Furthermore, we have updated reference 1 (World Health Organization). The changes to our reference list are marked in red on our revised manuscript.

Special thanks to you for your valuable comments.

Responses to the reviewers’ comments:

Reviewer #1:

Comment 1: I suggest an additional sentence to clarify the reason for exclusion of other types of depression being that you think these will interact with the estimated effects (?), which is not the case for co-morbid psychiatric disorders. In effect, you are conditioning by design, although that won't be clear to everyone.

Response 1: We sincerely appreciate the valuable comments. The revised texts read as follows:” To reduce clinical heterogeneity, we will exclude RCTs involving 20% or more participants with bipolar depression, treatment-resistant depression, psychotic depression, or peripartum depression, but not studies involving participants with other comorbid psychiatric disorders (e.g., anxiety disorder).” (Pages 6 to 7, lines 118 to 121)

Comment 2: I would keep an open mind to the possibility of splitting nodes according to levels of dose, because you may find adequate data with which to do this (more commonly we don't). As you have it written, defining multiple nodes for different doses of the same intervention would be a protocol deviation, which would be unfortunate. I suggest "We will consider different intensity levels or subtypes within the same type of non-pharmacological intervention as the same node in the network analysis in the first instance, or as different nodes if adequate data are available.”

Response 2: Thanks for your valuable comments. The revised texts read as follows:“We will consider different intensity levels or subtypes within the same type of non-pharmacological intervention as the same node in the network analysis in the first instance, or as different nodes if adequate data are available.” (Page 7, lines 128 to 131)

Comment 3: Okay, your reasons for using SMD are considered and reasonable. In that case, please bear in mind that the estimate of the heterogeneity variance will be critical to accurate interpretation of the results. You are assuming this parameter to be common across comparisons - that's reasonable and often the only option given the amount of data. That being so, if this estimate is not precise, it will be difficult to interpret the SMDs for the relative effects.

Response 3: Thank you for this valuable comment. Considering the importance of heterogeneity, we have added details in this regard. The revised texts read as follows:

“We will assume a common heterogeneity variance parameter (τ) for each network. We will compare the posterior distribution of the estimated heterogeneity with its empirical distributions to account for the magnitude of the estimated heterogeneity[1]. Specifically, we will use the first and third quantiles of their predictive distributions to characterize the amount of heterogeneity as low, moderate, or high[2].” (Page 14, lines 269 to 274)

“If we find significant heterogeneity and/or inconsistency, we will conduct subgroup analyses or network meta-regression analyses to explore the possible sources.” (Page 15, lines 302 to 303)

Comment 4: Thanks for making those updates regarding heterogeneity estimation and exploration. Strictly, I^2 is not a test, so please drop that word.

Response 4: We sincerely thank you for your careful reading. The revised texts read as follows:” Heterogeneity will be evaluated using Cochran’s Q test and I2.” (Page 12, line 244)

Comment 5: You might also note that you can compare the fit of the two models as an additional 'global' index of incoherence (although null difference is not evidence of coherence).

Response 5: Thanks for your valuable comments. We will calculate the posterior mean of residual deviance to access model fit. The closer the posterior mean of residual deviance is to the number of data points, the better the model fit[3]. (Page 13, lines 256 to 258)

Comment 6: I urge you strongly not to compare results of models with fixed or random effects terms for the treatment effects. Whereas with coherence we often do not know, in the case of whether treatment effects follow a distribution (random setting) or vary only due to sampling error (fixed setting), it is rare this is not known. Please choose an approach and stick with it, because the choice of term influences the ability to assess coherence. For example, in a network where there are few studies per comparison, and the model has a random effects term for treatment effects, there may be a false conclusion of no loop incoherence simply because the CIs are enlarged (and thus overlap for the indirect and direct estimates). Whilst, in this hypothetical scenario, the random effects term may be appropriate, there is the noted drawback of lesser ability to evaluate coherence. You won't have major problems, because you're using multiple means to evaluate coherence, but the point still stands that the treatment effects term needs to be chosen based on prior knowledge to avoid unwarranted model flexibility.

Response 6: We sincerely appreciate the valuable comments. The revised texts read as follows:” We will use the random-effects model in the Bayesian analyses.” (Page 13, lines 255 to 256) 

Comment 7: Please bear in mind the potential for results to vary between the Bayesian and frequentist settings. I suggest you also estimate treatment effects with mvmeta, compare them to those from OpenBUGS, and note the differences before looking at coherence metrics.

Response 7: Thanks for your valuable comments. The revised texts read as follows:” All analyses of the primary outcomes will be duplicated within a Frequentist framework using the ‘netmeta’ packages in R (version 4.1.2) software.” (Page 14, lines 274 to 275)

Comment 8: Lastly, please add a statement of where you will disseminate the study findings and data?

Response 8: Many thanks to your kind reminder. The revised texts read as follows:” This study will use secondary data and therefore does not require ethical approval. The results of this study will be disseminated through peer-reviewed publication.” (Page 3, lines 49 to 51)

Special thanks to you for your valuable comments.

Reviewer #3:

Comment 1: it would be grateful if they included a sentence in the methods on how to deal with missing data.

Response 1: Thanks for your valuable comments. When standard deviations (SDs) are not reported, we will first use the standard errors (SEs), t-statistics, p values, and so on to calculate the missing SDs. We will also contact the authors of the included study up to three times via email to obtain the relevant missing data and seek clarifications, which will be regarded as unobtainable if no clarification is provided within six weeks. (Page 11, lines 201 to 205)

Comment 2: what procedure to consider in the case of incomparable nodes of the interventions examined.

Response 2: Thank you for this valuable comment. We will only consider any non-pharmacological interventions recommended by current guidelines. To reduce inconsistency among trials, we will exclude studies that combined non-pharmacological interventions with specific drugs and studies using specific drugs as the comparator. We will consider different intensity levels or subtypes within the same type of non-pharmacological intervention as the same node in the network analysis in the first instance or as different nodes if adequate data are available. Furthermore, before running the network meta-analysis, we will evaluate the transitivity assumption (a fundamental assumption for the validity of indirect comparisons in network meta-analysis[4]) by comparing the distribution of potential effect modifiers across the different comparisons and evaluating consistency models. Important intransitivity might be manifested in the data as a statistical inconsistency (differences between direct and indirect evidence)[4, 5]. We will evaluate local and global inconsistency in Stata (version 15.1) software using the ‘network’ and ‘mvmeta’ packages. If there is no significant inconsistency, we will use the consistency model; otherwise, an inconsistency model will be used[6], and we will conduct sensitivity analyses for our primary outcomes by excluding local inconsistency spots to test the robustness of the results. (Page 7, lines 124 to 131; Page 14, lines 279 to 282; Page 15, lines 290 to 299; Page 16, lines 318 to 320)

 Comment 3: I also believe that the overall quality of evidence for each treatment comparison (on the main outcomes) should be done using the Confidence in Network Meta-Analysis (CINeMA) tool (https://cinema.ispm.unibe.ch).

Response 3: We sincerely appreciate the valuable comments. We will determine the overall quality of evidence for each treatment comparison using the Confidence in Network Meta­Analysis (CINeMA) tool (a web application based on the GRADE framework)[7, 8]. (Page 3, lines 44 to 45; Page 17, lines 337 to 398)

Special thanks to you for your valuable comments.

We are very grateful to you and the reviewers for your valuable comments on our paper. If there are any other modifications we could make, we would like to modify them further, and we appreciate your help. We hope that our manuscript can be considered for publication in your journal. Looking forward to hearing from you.

Thank you and best regards.

Yours sincerely,

Honghong Wang

Xiangya School of Nursing, Central South University, 172 Tongzuphill, Yuelu District, Changsha, Hunan Province, P. R. China, 410013.

Email: honghong_wang@hotmail.com

Reference

1. Turner RM, Davey J, Clarke MJ, Thompson SG, Higgins JP. Predicting the extent of heterogeneity in meta-analysis, using empirical data from the cochrane database of systematic reviews. Int J Epidemiol. 2012;41(3):818-27. Epub 2012/03/31. doi: 10.1093/ije/dys041. PubMed PMID: 22461129; PubMed Central PMCID: PMCPMC3396310.

2. Bighelli I, Rodolico A, García-Mieres H, Pitschel-Walz G, Hansen WP, Schneider-Thoma J, et al. Psychosocial and psychological interventions for relapse prevention in schizophrenia: a systematic review and network meta-analysis. Lancet Psychiatry. 2021;8(11):969-80. Epub 2021/10/16. doi: 10.1016/s2215-0366(21)00243-1. PubMed PMID: 34653393.

3. Dias S, Ades AE, Welton NJ, Jansen JP, Sutton AJ. Network meta-analysis for decision-making: John Wiley & Sons; 2018.

4. Higgins JPT TJ, Chandler J, Cumpston M, Li T, Page MJ, Welch VA (editors). Cochrane handbook for systematic reviews of interventions version 6.1 (updated September 2020). Cochrane2020. Available from: www.training.cochrane.org/handbook.

5. Wilson DB, Tanner-Smith E, Mavridis D. Network meta-analysis. Campbell Systematic Reviews. 2016;12(1):i-11. doi: https://doi.org/10.4073/cmpn.2016.1.

6. Dias S, Welton NJ, Caldwell DM, Ades AE. Checking consistency in mixed treatment comparison meta-analysis. Stat Med. 2010;29(7-8):932-44. Epub 2010/03/10. doi: 10.1002/sim.3767. PubMed PMID: 20213715.

7. Puhan MA, Schünemann HJ, Murad MH, Li T, Brignardello-Petersen R, Singh JA, et al. A GRADE working group approach for rating the quality of treatment effect estimates from network meta-analysis. BMJ. 2014;349:g5630. Epub 2014/09/26. doi: 10.1136/bmj.g5630. PubMed PMID: 25252733.

8. Nikolakopoulou A, Higgins JPT, Papakonstantinou T, Chaimani A, Del Giovane C, Egger M, et al. CINeMA: an approach for assessing confidence in the results of a network meta-analysis. PLoS Med. 2020;17(4):e1003082. Epub 2020/04/04. doi: 10.1371/journal.pmed.1003082. PubMed PMID: 32243458.

---

## [Decision Letter · Decision Letter 2]

7 Jun 2023

Comparative efficacy and acceptability of non-pharmacological interventions for depression among people living with HIV: A protocol for a systematic review and network meta-analysis

PONE-D-22-17925R2

Dear Author,

We’re pleased to inform you that your manuscript has been judged scientifically suitable for publication and will be formally accepted for publication once it meets all outstanding technical requirements.

Kind regards,

Guglielmo Campus, Ph.D DDS

Academic Editor

PLOS ONE

Additional Editor Comments (optional):

Reviewers' comments:

Reviewer's Responses to Questions

**Comments to the Author**

1. Does the manuscript provide a valid rationale for the proposed study, with clearly identified and justified research questions?

Reviewer #1: Yes

2. Is the protocol technically sound and planned in a manner that will lead to a meaningful outcome and allow testing the stated hypotheses?

Reviewer #1: Yes

3. Is the methodology feasible and described in sufficient detail to allow the work to be replicable?

Reviewer #1: Yes

4. Have the authors described where all data underlying the findings will be made available when the study is complete?

Reviewer #1: Yes

5. Is the manuscript presented in an intelligible fashion and written in standard English?

Reviewer #1: Yes

6. Review Comments to the Author

You may also provide optional suggestions and comments to authors that they might find helpful in planning their study.

Reviewer #1: Thank you for your responses to my comments. I am satisfied with each of them. Good luck with the study

7. PLOS authors have the option to publish the peer review history of their article (what does this mean?). If published, this will include your full peer review and any attached files.

Reviewer #1: **Yes: **Dr Matthew K Bagg

---

## [Editor Report · Acceptance letter]

19 Jun 2023

PONE-D-22-17925R2 

Comparative efficacy and acceptability of non-pharmacological interventions for depression among people living with HIV: A protocol for a systematic review and network meta-analysis 

Dear Dr. Wang:

I'm pleased to inform you that your manuscript has been deemed suitable for publication in PLOS ONE. Congratulations! Your manuscript is now with our production department. 

Kind regards, 

on behalf of

Prof. Dr. Guglielmo Campus 

Academic Editor

PLOS ONE